# Relevance of Nerve Biopsy in the Diagnosis of Chronic Inflammatory Demyelinating Polyneuropathy—*A Systematic Review*

**DOI:** 10.3390/diagnostics12071691

**Published:** 2022-07-11

**Authors:** Elena-Sonia Moise, Razvan Matei Bratu, Andreea Hanganu, Maria Sajin

**Affiliations:** 1Department of Pathology, Carol Davila University of Medicine and Pharmacy, 050474 Bucharest, Romania; elena.moise@drd.umfcd.ro (E.-S.M.); maria_sajin@yahoo.com (M.S.); 2Department of Anatomy, Carol Davila University of Medicine and Pharmacy, 050474 Bucharest, Romania; 3Department of Infectious Diseases, Carol Davila University of Medicine and Pharmacy, 050474 Bucharest, Romania; andreea.florea21@gmail.com; 4Department of Infectious Diseases, National Institute of Infectious Diseases Prof. Dr. Matei Bals, 021105 Bucharest, Romania; 5Department of Pathology, University Emergency Hospital, 050098 Bucharest, Romania

**Keywords:** chronic inflammatory demyelinating polyneuropathy, nerve biopsy

## Abstract

Chronic Inflammatory Demyelinating Polyneuropathy is an immune-mediated pathology of the peripheral nerves and nerve roots that leads to weakness and sensory symptoms. Given its clinical heterogeneity, often times diagnosis is challenging. Even though nerve conduction studies and clinical features are the main criteria used for diagnosis, supplementary investigations, such as nerve biopsies, cerebral spinal fluid examination and magnetic resonance studies, may be used in order to confirm the diagnosis. Given the fact that the hallmark in CIDP physiopathology is the demyelination process, nerve biopsies are used to demonstrate and assess the magnitude of the phenomenon. The question and the main interest of this review is whether histopathological findings are relevant for the diagnosis and can be useful in disease assessment.

## 1. Introduction

Chronic Inflammatory Demyelinating Polyneuropathy (CIDP) is an immune-mediated neuropathy with a heterogeneous clinical presentation consisting of a roughly symmetric involvement of peripheral nerves, which affects both motor and sensory components [1,2]. The classic clinical presentation of CIDP consists of a mainly motor and symmetric neuropathy, affecting both peripheral nerves and nerve roots, manifesting in proximal and distal muscle weakness with either a relapsing–remitting or a progressive course [3,4]. Therefore, even though CIDP can manifest in a heterogeneous manner, with a significant number of variants being described so far, the main electrophysiological and histopathological feature is segmental demyelination, which is the pathophysiological hallmark [1,5,6]. Typically, CIDP is responsive to glucocorticoid treatment, even though the therapeutic response might not be complete [7].

Typical electrophysiological findings in CIDP include partial conduction blocks, conduction velocity slowing consisting of prolonged distal motor latencies and delay or disappearance of F waves and temporal dispersion, as well as distance-dependent reduction of compound motor action potential (CMAP) amplitude [8].

Regarding neuropathological findings, CIDP is characterized by segmental demyelination and remyelination of peripheral nerves, a process leading to “onion bulb” formation—the term used to describe the aspect of axonal lesions when examined microscopically on transverse sections [3]. The demyelination process tends to occur in the proximity of Ranvier nodes, and it is uneven along the length of the nerve [9].

According to the EFNS/PNS Consensus Guidelines on the diagnosis and management of CIDP, revised in 2021, there are two types of CIDP classified according to the clinical features: typical CIDP and CIDP variants. The term “CIDP variants” replaced “atypical CIDP” from the EFNS/PNS Consensus Guidelines reviewed in 2010, as all phenotypes regarded as such in the 2010 Guidelines are now well characterized both clinically and electrophysiologically. The clinical criteria for CIDP are summarized in Table 1 [10].

The diagnostic of CIDP can be formulated solely based on clinical and electrophysiological criteria, with the latter having high sensitivity/specificity—95%/96% [11], 81%/96% [12], and 73%/91% [13] reported in different patient cohorts. Therefore, it is very accurate. However, according to 2021 EFNS/PNS Guidelines [10], the aim of this systematic review is to determine whether nerve biopsy is relevant for the diagnosis of CIDP and to allow pathologists and neurologists to deepen their understanding of the disease.

## 2. Materials and Methods

The research question was constructed using the Population/Intervention/Comparison/Outcome (PICO) format, as shown in Figure 1.

The Medline database was searched through its interface PubMed (1985 to May 2022) using the term “chronic inflammatory demyelinating polyneuropathy” combined, using the operator “AND”, with the term “nerve biopsy”. After applying the filters “full text”, “humans” and “English”, 174 articles were selected. Articles referring to pediatric population were excluded. The list of citations and bibliography of every identified article was further scanned for potentially relevant articles. Additionally, after a manual search was performed, we selected 5 more articles. Finally, 21 articles were included in this systematic review, having been published between 1998 and August 2020. The inclusion of publications was performed according to PRISMA 2020 system, as shown in Figure 2.

## 3. Results

The publications included in this systematic review are listed in Table 2.

The distribution of the selected studies, according to their design, is as follows: 52% retrospective, 33% reviews, 10% case control, and 5% case series.

We consider that the heterogeneity of the study designs is due to the relatively small number of studies conducted upon the subject.

### 3.1. Histological Findings

According to a review published 2012, by Peltier et al. [25], biopsy findings in CIDP are neither sensitive nor specific. They mainly consist of demyelination and mononuclear cell infiltration. Additionally, secondary axonal degeneration may be found, usually accompanied by clusters of regenerating fibers [1]. “Onion bulbs” are also a characteristic feature. Usually, they have a random pattern of distribution, lying among normally myelinated axons without “onion bulbs”. There are reasons to believe that the pattern of endoneurial perivascular macrophage clusters [20] and the extent of matrix metalloproteinase-9 immunoreactivity [34] can help with the differential diagnosis between inflammatory and non-inflammatory neuropathy. Immunoglobulin and complement deposits are rather common findings [26]. The histological findings are supported by the review published in 2014 by Mathey et al. [26], where immunopathogenesis of CIDP is summarized. The article presents the abiding theory of cell-mediated and humoral mechanisms involved in the pathogenesis of the disease.

A 2020 retrospective study conducted by Luigetti et al. [1], states that according to the American Academy of Neurology histopathologic criteria for CIDP, unquestionable proof of demyelination and remyelination needs to be present in more than five demyelinated fibers on electronic microscopy or there has to be evidence of demyelination/remyelination in at least 12% of 50 teased fibers, containing a minimum of four internodes each [35]. The main feature of CIDP is the evidence of macrophage-mediated demyelination, best diagnosed on electronic microscopy. Some studies demonstrate that very severe CIDP may be misdiagnosed as chronic idiopathic axonal neuropathy. In those cases, electronic microscopy can help differentiate demyelinating lesions [36].

In a case control study conducted in 2005 by Sommer et al. [20], the main goal was to establish whether macrophage clustering could be a useful marker in the histopathological diagnosis of CIDP. In order to fulfill this end-point, the study included 21 patients previously diagnosed with CIDP, who met the AAN criteria for CIDP, with the control group consisting of patients with hereditary polyneuropathies. The study shows that the number of perivascular macrophages per sections was higher in the CIDP group than in the hereditary polyneuropathy group, the difference being statistically significant. The article also states that, after having analyzed the data, when a cut off of 5% macrophage clusters is being met, CIDP can be predicted with a sensitivity of 66.7% and a specificity of 95%, when compared with hereditary polyneuropathies.

In 2010, Kulkarni et al. [23] published a retrospective study on 46 patients previously diagnosed with CIDP who underwent sural nerve biopsies. The article demonstrates that pathological findings consistent with the diagnosis of CIDP were observed in all patients. Four of the cases had all four histopathological characteristics; the authors analyzed subperineural edema, demyelination, “onion bulb” formation, and inflammation, 18 cases had any three of the pathological markers, 14 cases had any two of the features, and 10 cases had only one. An interesting observation made by the authors is that all cases demonstrated histopathological abnormalities, in contrast wtih electrophysiological alterations, which were demonstrated in 90.8% of the cases. The study also demonstrates that pathological findings might precede clinical and electrophysiological modifications.

In 2018, Min Xu et al. [28] published a case control study where they assessed the relevance of teased fiber analysis in patients previously diagnosed with neuropathies and included four new types of teased fibers (J-M). Nerve biopsies from 20 patients with CIDP were analyzed for the presence of teased fibers. The results demonstrated that in 12 cases of CIDP, teased fibers type J were found, yielding a specificity of 60%. Some of the characteristics of teased fibers type J, as described in the article, are: thick layers of collagen and Schwann cells covering the length of the fibers, forming a “rope-like” structure, loss of visibility of some of the Ranvier nodes and normal myelin, “onion bulb” formations visible on semi-thin toluidine blue sections, and loss of normal Schwann cell architecture. However, the authors state that they cannot conclude the sensitivity of the teased fibers, given the fact that the starting point of the study was to identify patients with a specific clinical–pathological diagnosis.

In 2019, Ikeda and colleagues [31] published a retrospective study carried out on 106 patients diagnosed with either typical CIDP or CIDP variants, according to the EFNS/PNS revised criteria. The study included 55 patients with typical CIDP, 15 patients with MADSAM (multifocal acquired demyelinating sensory and motor), 16 patients with DADS (distal acquired demyelinating symmetric polyneuropathy), and 15 patients with pure sensory polyneuropathy. The clinical-pathological correlations between phenotype and histopathological findings were as such: the typical CIDP group demonstrated no remarkable pathological features, and preferential involvement of the proximal and distal segments indicated a role of humoral immunity in the physiopathological process of the disease, at the site where the blood-nerve barrier is disrupted. By contrast, in all three subtypes of CIDP-variants: MADSAM, DADS, and pure sensory polyneuropathy, focal lesions were found, suggesting a shared physiopathological mechanism, different from the one of typical CIDP.

### 3.2. Value of Diagnosis

In an editorial published by Berini et al. [29] in 2019, the authors opine that nerve biopsy continues to be a useful diagnostic tool in diseases such as amyloid neuropathy, peripheral nerve sheath tumors, perineurioma, metastatic disease to nerve or neurolymphomatosis, inflammatory disorders (sarcoidosis or inflammatory demyelination such as atypical CIDP) and nerve vasculitis, where nerve biopsy continues to be the “gold standard” investigation.

Even though previous studies demonstrate a range of segmental de- and remyelination between 19–77% [17,37] in one study and 88% [17,38] in another, in patients previously diagnosed with CIDP, none of them compared those levels with the ones found in a control group with patients with no neuropathy, as did Bosboom et al. [17]. Therefore, by comparing the range of segmental de- and remyelination with the control group, no statistically significant difference was demonstrated. Moreover, neither segmental de- and remyelination, nor “onion bulb” formation or inflammatory infiltrates can help differentiate CIDP from diabetic polyneuropathy, as all three of those characteristics are present [17,39]. However, the “onion bulbs” formation in axonal polyneuropathies such as diabetic polyneuropathy, might represent “pseudo-onion bulbs” caused by a repeated anxonal degeneration and regeneration processes [40]. Probably the most surprising result of Bosboom et al. (2001) study is the lack of inflammatory features found in the CIDP group. The study tried to explain this fact, formulating three arguments: 1. The physiopathological changes of peripheral nerves in CIDP usually occur in proximal portions of the nerve, which are far less accessible of biopsy; 2. Given the fact that the sural nerve is a sensitive nerve, it is less prone to carry characteristic lesions, as motor signs and symptoms are more prominent than sensory ones [3,17]; and 3. By the time the biopsy is performed, the physiopathological processes might have ceased [17].

According to a case series published by Boukhris et al. [19] in 2004, all eight patients included in the study had clinical features compatible with CIDP; however, they were without electrophysiological criteria. The nerve biopsy findings—features of demyelination, naked axons without myelin sheath, or with sheaths that were too thin compared to the axonal dimensions—were compatible with the biopsy findings of patients who met the clinical and electrophysiological criteria. Given these observations, the authors raise the question whether histopathological criteria should be used in selected cases, in order to establish a diagnosis of certainty.

In 1998, Molenaar et al. [14] conducted a study designed to assess whether nerve biopsies performed on patients who had already met the clinical and electrophysiological criteria for CIDP would increase the sensibility of the diagnosis. In order to complete this, a neurologist experienced in peripheral nerve disorders reviewed clinical data and pathology results of the patients included in the study. Based on the results of the assessment, the neurologist was able to discriminate between the CIDP group and the non-CIDP group without being influenced by the biopsy result in a statistically relevant manner. Taking in consideration the pathology report, the decision was changed in five patients, two of whom had other diagnoses. Therefore, the conclusion of the study was that nerve biopsies should not be among diagnostic criteria for CIDP.

In 2000, Haq, RU et al. [16] conducted a retrospective study on 24 patients previously diagnosed with CIDP who had undergone nerve biopsy. Among the results, the study demonstrated a lack of correlation between electrophysiological and histological criteria in patients with CIDP, demonstrating that slow motor conduction did not predict de- or remyelination processes identified by teased fiber analysis in sural nerves. This observation had already been made in a 54-patient retrospective study, conducted by Barohn et al. [41]. Furthermore, the study demonstrated a statistically significant similarity between the American Academy of Neurology (AAN) criteria for teased fiber analysis and the electrophysiological criteria used, regarding the demyelination in CIDP. Moreover, the study demonstrates that histological markers of demyelination are present, regardless of its evidence in electrophysiological studies. The authors comment on the fact that their results contradict the Molenaar et al. study conducted on 23 patients [14], in which no significant added value for the diagnosis of CIDP was brought by the nerve biopsy. Haq et al. states that the difference in results resides on several arguments: firstly, the study included a large proportion of patients who only met a certain set of electrophysiological criteria; secondly, the nerve biopsies collected from the patients were not analyzed with electronic microscopy; and thirdly, there was a lack of quantitative pathologic conditions in nerves. Regarding electronic microscopy, the study demonstrates that the American Academy of Neurology criteria are almost twice as sensitive as the electrophysiological ones.

In an article published in 2003 by Vallat et al. [18], the diagnostic value of the nerve biopsy in CIDP variants was studied. All eight patients included in the case series had atypical clinical signs: almost exclusively sensory symptoms, asymmetry, central involvement associated with peripheral signs, and predominantly distal involvement. However, given the heterogeneity of CIDP, none of the clinical presentations excluded the disease [39]. The study demonstrated that, given the atypical form of manifestation, electrophysiological studies were equivocal, and that nerve biopsy was crucial in demonstrating demyelination. The authors state that, as the disease progresses, so do inflammatory demyelinating lesions, subsequently leading to axonal damage. According to the authors, this is the reason why several axonal neuropathies respond to immunosuppressive therapy. In conclusion, the article draws attention to the fact that nerve biopsy may lead to the diagnosis of CIDP in chronic neuropathies of apparently unknown cause, with an atypical form of clinical presentation.

In a systematic review performed by Sommer et al. [22] in 2010, an attempt to answer a series of questions regarding nerve biopsies was made. The main topics taken into consideration were the methodology of performing a nerve biopsy, tissue processing, diagnostic usefulness of paraffin histology, plastic-embedded sections, frozen sections, imunohistochemistry, inflammatory cell detection, teased fiber studies, electron microscopy, special markers, morphometry, and the comparison of histopathological makers with seric and cerebrospinal fluid markers. Regarding the methodology of performing a nerve biopsy, the authors state that the procedure should be performed by a trained medical professional, under sterile conditions, and with local anesthesia. However, according to the review, no studies addressed the question about the choice of nerve to be biopsied and the surgical techniques. When evaluating the usefulness of teased fiber analysis, the review demonstrates that when analyzing the results published by Haq et al. [16], Bosboom et al. [17], and Deprez et al. [42], the diagnostic relevance of the procedure remains unclear. For example, 7 out of 14 patients included in the study conducted by Haq et al. [16], who fulfilled teased fiber analyses criteria for demyelination, did not meet the electrophysiological criteria. However, three of them responded to immunosuppressive treatment. Furthermore, after studying data from 21 patients with CIDP, Bosboom and colleagues [17] came to the conclusion that teased fiber analysis was not useful when trying to differentiate CIDP from chronic idiopathic axonal neuropathy.

In 2019, Nathani et al. [30] conducted a systematic review that raises awareness regarding the appropriate circumstances in which a nerve biopsy should be performed. The article states a series of conditions in which nerve biopsy may be a diagnostic tool. Among those, there are: vasculitic neuropathies with no evidence of extraneural vasculitis and no response to immunosuppressive treatment, neurolymphomatosis which could not have otherwise been confirmed, primary nerve/nerve sheath tumors, pure neurotic leprosy, amyloid neuropathy, sarcoid peripheral neuropathy, with no evidence of extraneural involvement, IgG4-related disease/neuropathy, para-proteinemic neuropathy, hereditary neuropathy, and CIDP variants non-responsive to immunosuppressive treatment. The study also states that in a retrospective study carried out on 146 patients with definite CIDP according to EFNS/PNS 2006 criteria, supporting diagnostic elements, including nerve biopsies, were required for 25% of them [30,43].

In a retrospective study published in 1999 by Vital et al. [15], 18 patients previously diagnosed with chronic inflammatory demyelinating polyneuropathy associated with dysglobulinemia were studied. The dysglobulinemic status of the patients was discovered during the medical evaluation of the clinical presentation. IgG monoclonal gammopathy was present in eight cases, IgM monoclonal gammopathy was discovered in another eight cases, one of the patients had IgG-IgM biclonal gammopathy and the last one had IgM monoclonal cryoglobulinemia. All eight patients diagnosed with IgM monoclonal gammopathy and the patient with IgG-IgM biclonal gammopathy had anti-MAG antibodies present in the serum. They all underwent superficial peroneal nerve biopsies. The histopathological lesions found in the IgG monoclonal gammopathy cases are as follows: no vasculitis traits were found in any of the nerve specimens examined, a major loss of myelinated fibers was revealed, all exhibiting association of demyelination and axonal lesions at the ultrastructural level, half of the patients had macrophage-associated demyelination and seven patients had “onion bulb” lesions in both myelinated and demyelinated fibers. In the IgM monoclonal gammopathy group, among the histopathological findings were: various degrees of loss of myelinated fibers, association of demyelination and axonal lesions revealed through electronic microscopy in all eight cases, macrophage-associated demyelination in six cases, segmental demyelination and “onion bulb” formation in all eight cases. The IgG-IgM biclonal gammopathy associated moderate loss of myelinated fibers, no macrophage-associated demyelination, axonal lesions in both myelinated and demyelinated fibers, and “onion bulb” formations. In the IgM monoclonal cryoglobulinemia case, the nerve biopsy revealed no significant loss of myelinated fibers, marked macrophage-associated demyelination, segmental demyelination, and “onion bulb” formation, and the coexisting axonal damage was moderate. The article demonstrated that a number of histopatological features are related to the dysglobulinemic status, such as macrophage-associated demyelination, which was found in 11 of the 18 cases. The study also found that no specific morphopathological features were exhibited in the IgG monoclonal gammopathy group and in the IgM monoclonal cryoglobulinemia case. However, the IgM monoclonal gammopathy group and the IgG-IgM biclonal gammopathy case, which all had in common serum anti-MAG activity, demonstrated a widening of the outermost myelin lamellae.

In a case series conducted by Mathis et al. [24] in 2012, the role of nerve biopsy in the differential diagnosis of CIDP is highlighted. The article presents the cases of five patients who fulfilled EFNS/PNS criteria for CIDP at the time of publication. However, when they underwent nerve biopsy, pathological features consistent with the diagnosis of amyloid neuropathy were observed (amorphous deposits of various sizes positive for the Congo red stain). Genetic testing demonstrated a V30M mutation in the TTR gene, confirming the hereditary amyloid neuropathy diagnosis in three of the cases, whereas the other two patients were confirmed with primary amyloid polyneuropathy.

A retrospective data review published in 2018, by Allen et al. [27] included 65 patients that were previously diagnosed with CIDP. The study was meant to assess the adherence of medical professionals to EFNS/PNS diagnostic criteria. After having analyzed the data provided, the authors reached the conclusion that only 11% of the patients had sufficient criteria to confirm a CIDP diagnosis. Out of the 65 patients, only three of them had undergone nerve biopsy, raising the question whether if histological assessment was to be made, the rate of misdiagnosis might have been lower.

In a review published in 2020 by Stino et al. [33], misdiagnosis of CIDP is also evaluated. The main causes that lead to this phenomenon, according to the authors, include amplitude-dependent slowing in length-dependent neuropathies, amplitude-independent slowing in diabetic polyneuropathies, isolated distal latency changes in the fibular nerve (when recording over the extensor digitorum brevis), and focal slowing across common entrapment sites [44].

### 3.3. Complications of Nerve Biopsy

A retrospective study carried out by Hilton et al. [21] on 50 patients who underwent sural or peroneal nerve biopsies in order to establish a diagnosis for the clinical tableau compatible with neuropathy, demonstrated that 33 of them were diagnosed with axonal polyneuropathy without specific features, 6 with peripheral nerve vasculitis, 3 with chronic inflammatory demyelinating polyneuropathy, 2 with paraproteinaemic polyneuropathy, 2 with motor neuron disease, 1 with X-linked spinobulbar atrophy, 1 with familial neuropathy, and in 2 of the cases, the diagnosis remained uncertain. The study also demonstrated various neurological and surgical complications arising from the procedure and compared their frequency both in sural and peroneal nerve biopsies. Thus, according to Hilton et al, the most frequent neurological complications were postoperative pain, dysesthesia, and paraesthesia. In the sural nerve biopsy group, the incidence of postoperative pain was 29%, that of dysesthesia was 29%, and paraesthesia was reported by 38% of the patients during the follow up period. In the peroneal nerve biopsy group, the incidence of postoperative pain was 15%, that of dysesthesia was 11%, and that of paraesthesia was 54%. Another interesting observation of the study is that the incidence of neurological complications was greater in the group of patients without preoperative sensory symptoms, 23% of whom reported dysesthesia, 66% paraesthesia, and 30% postoperative pain, compared with the group of patients with preoperative sensory symptoms, of whom 15% reported increased dysesthesia, 15% paraesthesia and 15% postoperative pain.

Regarding surgical complications, the study demonstrated that delay in wound healing, wound infections, and hematomas were the most frequently reported, as follows—nine patients reported delay in wound healing, the cut off being established at 3 weeks, four patients reported infection, and two patients reported hematomas at the biopsy site. The study mentions that from the nine patients reporting delay in wound healing, five were treated with immunosuppressants and one was previously diagnosed with diabetes.

The study also evaluated the use of fascicular biopsy—a more targeted form of nerve biopsy—and came to the conclusion that it is not successful in reducing postoperative complications [21,45]. Immobilization of the leg in a plaster cast 7 days after the procedure, in order to diminish wound healing delay, was a procedure worthy of mention by the article [21,46]. Microsurgical repair of the biopsied nerve was as well; however, this procedure implies that no more than one centimeter of the nerve is to be removed [21,47].

The results arising from the study are consistent with previous research, as the authors state [20]. The cumulative data analyzed in the article show a 30% incidence of postoperative pain, 33% of dysesthesia [21,45,48,49,50,51,52,53], 40% of paraesthesia [21,48,49,53], and 8% of wound infection [21,51,52,53,54]. However, we mention that the data shown by the study were not statistically analyzed, probably due to the small number of patients, as the article mentions [21]; thus, it is impossible to state that it is statistically relevant.

## 4. Discussions

CIDP is a rare form of polyneuropathy, but has a great impact on the daily living of patients, considering that it is a disability that arises from its manifestations. Given the treatable nature of the disease, an unequivocal diagnosis is mandatory. Therefore, the American Academy of Neurology and the European Federation of Neurological Societies/Peripheral Nerve Society continue to revise and improve their guidelines [10,55] in order to elaborate diagnosis criteria, with greater sensibility and specificity. However, considering the heterogeneity of the condition, there are cases that do not meet the clinical and electrophysiological criteria, and need adjacent criteria in order to formulate a definite diagnosis. As the EFNS/PNS Guideline 2021 [10] states, nerve biopsy can represent a supportive criterion, but the task force suggest performing it only under certain circumstances.

Taking into account the findings published by Sommer et al. [20] in 2005, it is clear that using histopathological markers in the diagnosis of CIDP may be an efficient way to increase the sensibility and specificity of nerve biopsy as a diagnostic tool. As the aforementioned study demonstrates, the use of macrophage clustering may be a viable way to differentiate CIDP from hereditary polyneuropathies. However, the article shows that when compared with other inflammatory polyneuropathies, the specificity of macrophage clustering decreases significantly.

Maybe one of the most promising values of nerve biopsy in CIDP is the presence of immunohistochemical markers detected on nerve biopsy specimens, used to increase the specificity of the diagnostic tool. As Sommer and colleagues [22] show in the systematic review performed in 2010, T-cell counts show diagnostic efficacy for CIDP; however, the observation is made taking into consideration a series of retrospective studies conducted by Schmidt et al. [56] and Kiefer et al. [57]. In our opinion, the usefulness of immunohistochemistry markers should be further analyzed in prospective studies, which are unavailable at this moment. We also mention the relevance of teased fiber analysis reviewed by Sommer et al. [22] upon which the article came to a conclusion that remains unclear. However, a retrospective study published in 2018 by Min Xu and colleagues [28] demonstrated the presence of four more types of teased fibers, one of which (type J) was found in patients previously diagnosed with CIDP with a specificity of 60%.

In our opinion, nerve biopsy might prove itself useful in the differentiation between relapsing–remitting and progressive forms of CIDP. As demonstrated in the study published by Kulkarni and colleagues [23], endoneurial inflammation and demyelination were features more commonly associated with the relapsing–remitting form, whereas perivascular epineurial inflammation was found in all cases in which a progressive form of CIDP was diagnosed. Given this observation, we opine that prospective studies would be useful in order to assess the true scientific value of this information, having taken into consideration the poor response to the immunosuppressive treatment of progressive CIDP. Moreover, they would help determine whether histopathological findings can bring useful information, especially when electrophysiological studies in patients with relevant clinical signs and symptoms are normal. Additionally, future studies that analyze the ability of ultrasonography or MRI to help detect the best nerve or nerve segment for histopathological analysis would be useful as well.

Taking into consideration the results of the retrospective data review published by Allen et al. [27] in 2018, the question arises whether the misdiagnosis of CIDP may be avoided by performing nerve biopsies more often. Moreover, we consider the role of nerve biopsy is highly important in the differential diagnosis of CIDP, helping medical professionals differentiate it from CIDP-mimics such as amyloid neuropathy. This observation is supported by both the article published in 2012 by Mathis et al. [24], where the role of nerve biopsy in diagnosing amyloid neuropathy in patients who fulfilled the EFNS/PNS clinical and electrophysiological criteria for CIDP is being portrayed and in the study conducted by Sommer and colleagues [20] in 2005, where it is demonstrated that macrophage clustering may help differentiate between inflammatory and hereditary neuropathies.

Regarding future research themes, in our opinion, prospective studies would be useful in determining whether histopathological findings can bring significant information, especially when the electrophysiological studies in patients with relevant clinical signs and symptoms are normal. Additionally, future studies that analyze the ability of ultrasonography or MRI to help detect the best nerve or nerve segment for histopathological analysis would be useful as well.

Formulating a clear and irrefutable CIDP diagnosis is crucial, given the fact that the condition is treatable. Among various treatments, corticosteroids and intravenous immunoglobulin treatments have been demonstrated to be efficient in treating the condition both in the short and long term [24,58]. Therefore, therapeutic response represents a useful criterion for the diagnosis of CIDP [24,59] and the lack of response to immunomodulatory drugs may represent the indication for a nerve biopsy to establish whether the clinical features of the case may be attributed to a CIDP-mimic [2,24].

Having taken all of the above into account, and after having analyzed all the studies included in this systematic review, we consider that nerve biopsy may be a useful tool in the diagnosis of CIDP variants; however, given that it is a highly invasive procedure, and its significant rate of complications [21], it should be performed only in selected cases, when its diagnostic significance can lead to a better understanding of the particular case.

## Figures and Tables

**Figure 1 diagnostics-12-01691-f001:**
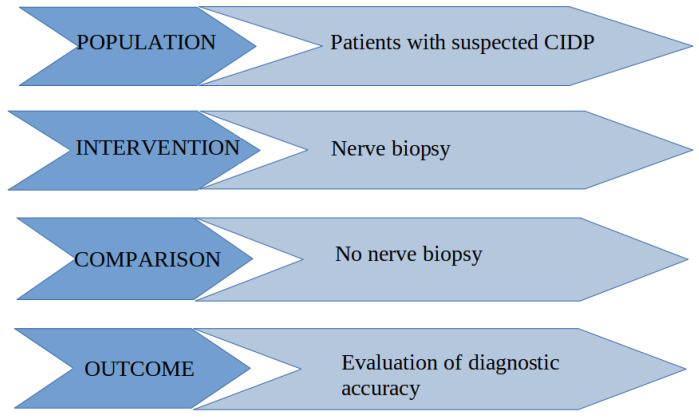
Construction of the research question using the PICO format.

**Figure 2 diagnostics-12-01691-f002:**
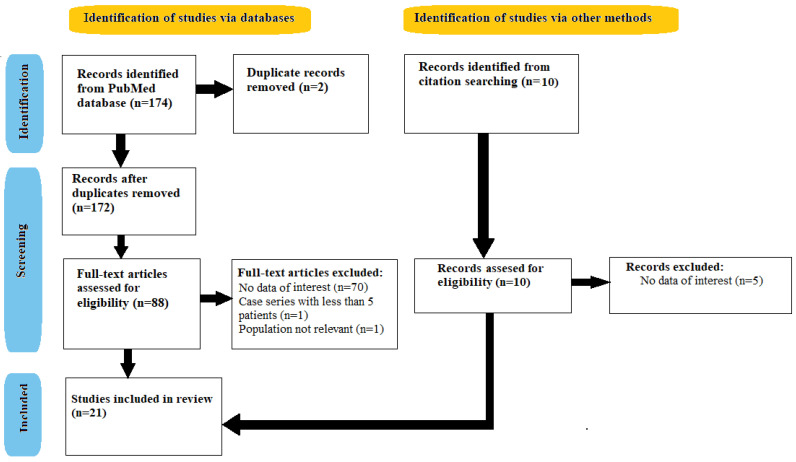
Selection criteria (PRISMA 2020 Flow Diagram).

**Table 1 diagnostics-12-01691-t001:** Clinical criteria for CIDP [10].

Typical CIDP	CIDP Variants
All of the following:	One of the following, but otherwise as in typical CIDP (tendon reflexes may be normal in unaffected limbs)
Progressive or relapsing, symmetric, proximal and distal muscle weakness of upper and lower limbs, and sensory involvement of at least two limbs	Distal CIDP: distal sensory loss and muscle weakness predominantly in lower limbs
Progressing over at least 8 weeks	Multifocal CIDP: sensory loss and muscle weakness in a multifocal pattern, usually asymmetric, upper limb predominant, in more than one limb
Absent or reduced tendon reflexes in all limbs	Focal CIDP: sensory loss and muscle weakness in only one limb
	Motor CIDP: motor symptoms and signs without sensory involvement
	Sensory CIDP: sensory symptoms and signs without motor involvement

**Table 2 diagnostics-12-01691-t002:** Articles included in this review.

No.	First Author’s Name	Year of Publication	Study Type
1.	Molenaar, DMS [14]	1998	Retrospective
2.	Vital, A [15]	1999	Retrospective
3.	Haq, RU [16]	2000	Retrospective
4.	Bosboom, W.M.J. [17]	2001	Case Control
5.	Vallat, J [18]	2003	Case series
6.	Boukhris, S [19]	2003	Retrospective
7.	Sommer, C [20]	2005	Case Control
8.	Hilton, D [21]	2007	Retrospective
9.	Sommer, C [22]	2010	Systematic Review
10.	Kulkarni, G [23]	2010	Retrospective
11.	Mathis, S [24]	2011	Retrospective
12.	Peltier, A [25]	2012	Review
13.	Mathey, E [26]	2015	Review
14.	Allen, J [27]	2017	Retrospective
15.	Min Xu [28]	2018	Case Control
16.	Berini, S [29]	2019	Review
17.	Nathani, D [30]	2019	Systematic Review
18.	Ikeda, S [31]	2019	Retrospective
19.	Eftimov, F [32]	2020	Review
20.	Luigetti, M [1]	2020	Retrospective
21.	Stino, A [33]	2020	Review

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
