# Peer review of "Relevance of Nerve Biopsy in the Diagnosis of Chronic Inflammatory Demyelinating Polyneuropathy—*A Systematic Review"

_diagnostics, 2022, doi:10.3390/diagnostics12071691_

Round 1

Reviewer 1 Report

Authors clearly addressed all the most recent data avalaible about the usefulness of nerve biopsy in CIDP diagnosis. References are up to date. I would add, if available, data of nerve biopsy performed and its results in larger CIDP databases/registries.

Author Response

Firstly, we would like to thank the reviewer for taking the time to review our manuscript. According to their suggestion, we added a summery of the article based on the study with the largest patient cohort undergoing histological assessment of nerve biopsy, published up to date, to the best of our knowledge. The exact location in the manuscript is: pages 5-6, lines 143-155.

Reviewer 2 Report

 Due the relevance of the study . It isimportant to discuss a little more or to propose in which cases the biopsy could be done

Author Response

We thank the reviewer for taking the time to revise our manuscript and for pointing this out. We have discussed more in depth both the purpose of nerve biopsy and the benefits it adds to the diagnosis of CIDP, based on the studies included in our systematic review.

Reviewer 3 Report

First, this manuscript was not written in a format of review. It was in a format of a standard research article "introduction; methods; results; discussion". As a review, this manuscript just listed the published paper with a majority of the content from the literature but without an in-depth summary and discussion. No logical structure. No useful information.

Author Response

We appreciate the reviewer’s suggestion and agree that it would be useful to deepen our discussion and summary based on the literature we have included in our review. As such, we added more information regarding the studies that we cited and provided our point of view on the usefulness of nerve biopsy in the “discussions” section. We hope that by adding more information in a structured manner, the logic and the relevance of the article has improved. Regarding the structure of the manuscript, we tried to check all the points on the PRISMA guidelines checklist (required by the publication when elaborating a systematic review), which includes the following sections: "title", "abstract", "introduction", "methods", "results", "discussion" and "other information".

Round 2

Reviewer 3 Report

The authors made considerable improvements to the manuscript. The current version is easier to follow.